# Diagnostic Framework of Pelvic Massive Necrosis with Peritonitis following Chemoradiation for Locally Advanced Cervical Cancer: When Is the Surgery Not Demandable? A Case Report and Literature Review

**DOI:** 10.3390/diagnostics12020440

**Published:** 2022-02-09

**Authors:** Elisabetta Sanna, Giacomo Chiappe, Fabrizio Lavra, Sonia Nemolato, Sara Oppi, Antonio Macciò, Clelia Madeddu

**Affiliations:** 1Department of Gynecologic Oncology, A. Businco Hospital, ARNAS G. Brotzu, 09100 Cagliari, Italy; dr.elisabettasanna@gmail.com (E.S.); giacomo.chiappe@aob.it (G.C.); fabrizio.lavra@aob.it (F.L.); 2Department of Pathology, ARNAS G. Brotzu, 09100 Cagliari, Italy; sonia.nemolato@aob.it; 3Hematology and Transplant Center, A. Businco Hospital, ARNAS G. Brotzu, 09100 Cagliari, Italy; sara.oppi@aob.it; 4Department of Surgical Sciences, University of Cagliari, 09100 Cagliari, Italy; 5Department of Medical Sciences and Public Health, University of Cagliari, 09100 Cagliari, Italy; clelia.madeddu@tiscali.it

**Keywords:** cervical cancer, radiotherapy, pelvic necrosis, sepsis, laparoscopic surgery, radiotherapy-induced complications

## Abstract

Concurrent platinum-based chemoradiation (CCRT) is the established treatment for locally advanced cervical cancer and has an acceptable toxicity. Radiation-induced necrosis of the uterus and pelvic tissue is a rare and usually late potential complication. Limited data are available about its management. Here, we describe a case of a patient affected by a locally advanced cervical cancer (stage IVA) who received CCRT, obtaining a partial response with persistence of bladder and rectal infiltration. Unfortunately, after the first brachytherapy dose, the patient developed a worsening clinical picture with fever and altered laboratory data indicative of sepsis; the computed tomography revealed a massive necrosis of the uterus with pelvic abscess and peritonitis. We performed a laparoscopic emergency surgery with removal of the necrotic tissue, supracervical hysterectomy, bilateral-oophorectomy, and abscess drainage. Thereafter, once the severe inflammatory condition was resolved, the patient underwent pelvic exenteration with palliative/curative intent. The postoperative PET/CT was negative for residual disease. However, the patient needed further hospitalization for re-occurrence of peritonitis with multiple abscesses. A careful diagnosis is crucial in locally advanced cervical cancer patients who, after CCRT, present persistent pain and problematic findings at imaging and laboratory parameters. In these cases, radiation-induced necrosis of the pelvis should be suspected. This case helps to clarify the central role of surgery, especially when actinic necrosis leads to complications such as abscess, fistulae, and extensive tissue destruction that cannot be conservatively medically handled. Laparoscopy represents an ideal approach to realizing the correct diagnosis, as well as enabling the performance of important therapeutic surgical procedures.

## 1. Introduction

Concurrent platinum-based chemoradiation (CCRT) is the standard treatment for locally advanced cervical cancer and can be performed with acceptable toxicity [1,2]. However, complications can occur during and after treatment, involving the urinary tract, rectum, vagina, and small intestine. Notable side effects include radiation enteritis, genito-intestinal and genito-urinary fistula, urinary tract obstruction, bleeding, shortening of the vagina, and, rarely, radiation necrosis of pelvic tissues [3]. While gastrointestinal and urinary toxicity is well documented [3], minimal data concerning radiation necrosis and its management are available. Radiation necrosis, a focal structural lesion that occurs at the original tumor site, is a potential long-term complication of radiotherapy. This report describes a case of cervical cancer treated using CCRT that showed partial response and persistence of bladder infiltration. Unfortunately, the patient developed massive necrosis with pelvic abscess and peritonitis. We opted for a laparoscopic interval surgery followed by a palliative pelvic exenteration with curative aspirations.

## 2. Case Report

A 70-year-old female patient was diagnosed with FIGO stage IVA cervical cancer that had spread to the right ovary, bladder, right ureter, superior half of the vagina, and pelvic lymph nodes. The patient underwent a right nephrostomy because of right ureteral infiltration and associated hydronephrosis. The patient received CCRT with weekly cisplatin (40 mg/m^2^) for five cycles, whole pelvic radiation 45 Gy in 25 fractions over 5 weeks, boost 12.6 Gy in seven fractions, and brachytherapy 7 Gy. Following the first brachytherapy administration, the patient presented with a fever (>39 °C) and was admitted to our Unit of Gynecologic Oncology at Businco Hospital, ARNAS G. Brotzu, Cagliari. Laboratory tests showed an increase in white blood cells (WBC) (20,800 U/L), platelets (566,000 U/L), C-reactive protein (CRP) (23.1 mg/dL), and fibrinogen (844 mg/dL), and decreased hemoglobin level (9.6 g/dL). A computed tomography (CT) scan was performed, which showed the presence of a cervical tumor with extensive necrosis (78 × 43 mm), infiltrating the bladder and without cleavage plan with the rectum. Adjacent to the tumoral mass, a large (multichambered) abscess measuring 124 × 85 × 125 mm was observed which extended to the right parietocolic wall (Figure 1). No evidence of distant metastases was observed.

In consideration of the preoperative findings, the patient underwent emergency surgical intervention. We opted for a laparoscopic surgery, where the necrosis of the uterus, adnexa, and right parametrium and the overlying abscess were macroscopically visible. We performed drainage and washing of the pelvic cavity with removal of abscesses and of the necrotic tissue that included the body of the uterus and adnexa, then carried out supracervical-hysterectomy, and bilateral salpingo-oophorectomy. The radical surgery was postponed until the inflammatory parameters stabilized. The patient made a good recovery postoperatively, highlighted by a rapid decrease of inflammatory indexes and stable parameters. The patient was discharged on day seven. The definitive histopathological examination revealed persistency of adenocarcinoma in the myometrium. The uterine serosa, parametrium, and adnexa were tumor-free. In detail, the lower surgical margin (isthmic) was necrotic with inflammation, without presence of tumoral cells; the fundic uterine margin showed active inflammation without tumoral localization; the tubaric surgical margin was involved by active inflammation without evidence of neoplastic cells. Actinic necrosis was observed in the pelvic abscess (Figure 2).

After 30 days, once the severe inflammatory picture was solved, the patient underwent radical laparotomic surgery with palliative/therapeutic intent. The surgery was performed with radical cystectomy en bloc with the cervix. The small intestine with the results of actinic necrosis showed a sub-occlusive pattern, necessitating a large ileal resection. Rectal resection and termino-terminal sigmorectal anastomosis and a preventive colostomy were performed due to the suspected rectal infiltration and absence of surgical cleavage plane from the cervix. Because of the ileal resection, ileal conduit urinary diversion according to the Bricker technique was contraindicated; therefore, we opted for a left ureterocutaneostomy [4,5]. In addition, the right ureter showed neoplastic infiltration in the inferior third, and therefore, we removed the infiltrated tissue and closed the ureter (as the patient had already undergone a right nephrostomy). The final histopathological examination showed: the cervical structures replaced by necrotic tissue, steatonecrosis with acute suppurated inflammation, leukocyte fibrin tissue, sclerosis, and hyalinosis of the vessels with no evidence of neoplastic residues; foci of adenocarcinoma infiltrating at full thickness the bladder muscle coat until the mucosa; extensive necrosis involving the small intestine, rectum, appendix, and peritoneum. The cervical surgical margin presented active inflammation without any localization of neoplasia; vesical surgical margins were free of neoplastic cells with serositis; the rectal surgical margin showed active inflammation without neoplastic localization; the serosal of the colon was covered with active inflammation and necrosis, free of tumoral nest. Pathological stage confirmed involvement of the bladder (pT4a). The patient had an uneventful recovery, and she was discharged on day 12. After one month the patient underwent a postoperative PET/CT that resulted negative for relapsed/residual disease. However, afterward the patient needed hospitalization for reappearance of a widespread peritonitis with multiple abscess collections, which were treated with drainage and antibiotics therapy. The patient is currently alive and functioning with a left ureterocutaneostomy, right nephrostomy, and colostomy.

The study was carried out according to the guidelines of the Declaration of Helsinki. According to the Italian Regulatory Agency for observational trials not involving drugs, the protocol was disclosed to the Institutional Ethics Committee of “Azienda Ospedaliero Universitaria di Cagliari”, Cagliari, Italy. The patient signed an informed consent form for the surgical and medical treatment. Written informed consent has been obtained from the patient to publish this paper.

## 3. Discussion

The efficacy of concurrent chemoradiation for treating advanced cervical cancer has been widely reported [6,7].

Acute and late toxicity have been described as radiation enteritis, genito-intestinal, genito-vesical, and vesico-intestinal fistula, urinary tract obstruction/stenosis, fibrosis, and necrosis [3,8].

Nakano et al. examined 1148 patients treated with radiotherapy alone. They reported an incidence of late complications in the rectum, small intestine, and urinary tract at 10 years as 22%, 9%, and 18%, respectively, with significant complications occurring in 4.4%, 3.3%, and 0.9% of cases, respectively [9]. Yamada et al. [10] retrospectively analyzed late adverse events and their chronological appearance by comparing patients treated by CCRT with those who underwent RT alone. No significant differences between the two groups were reported, suggesting that concomitant CCRT does not increase the frequency and severity of late complications.

Toxicity may depend on the radiation dose, treatment volume, and tissue health within the target area. Additionally, different tissue types may develop sequelae following RT at varying rates and timing of onset [11]. Pelvic radiation doses greater than 45 Gy are associated with significantly higher complication rates [10]. The use of combined modalities, such as surgery plus chemoradiation, has been identified as a major risk factor for late complications [12].

Moreover, of note, tumor extension at diagnosis has been correlated with increased incidence of complications, such as fistulae (recto-vaginal and vesico-vaginal fistulae), which represent a therapeutic challenge in very locally advanced cervical cancers [13,14,15]. Therefore, the use of CCRT in bulky and very advanced cervical cancer, as in our case, has major risk. In particular, it is often reported that bladder involvement and infiltration is associated with a strong fragility of vesical wall after radiation that almost constantly favors complications related to radiotherapy with consequent fistulae formation [14,16,17]. In addition, surgery after CCRT in such an advanced stage of disease may be related to more frequent late complications [18].

Necrosis of pelvic tissues represents a rare, usually late, complication of CCRT. Risk factors include total dose of radiation therapy, the radiation therapy field size (likely as a function of disease extension), advanced stage disease, history of abdominal surgery (with compromised vascular supply), and diabetes [19]. The pathophysiology of pelvic necrosis is characterized by cellular depletion and tissue devascularization. Compromised blood flow causes secondary hypoxia within the ulcerated area and can lead to fistulae development [11]. Delayed or chronic ulcers that form months to years after RT are a consequence of both epithelial atrophy as well as hypovascularity. Chronic ischemia severely impairs wound-healing capacity and makes the area susceptible to infection. Additionally, there is an increased risk for an underlying superinfection, given the potential fecal contamination of the distal vagina and the fact that necrotic tissue is a source of nutrients for bacteria [19]. The early onset of necrosis (as in our patient) could be attributed also to an individual-specific increased sensitivity to radiotherapy. In this regard, it has been reported that anemia, oxygenation status, erythropoietin, and endogenous markers of tumor hypoxia may influence tumor control and the radiation sensitivity of healthy tissues [11]. In our particular case, the patient had a very locally advanced disease, notable for involvement of the bladder and rectum, and, as such, fistulous communication between these organs was an expectation of therapeutic effect. This may have favored severe infection and abscess formation which likely precipitated the normal tissue necrosis.

Pelvic necrosis following CCRT for cervical cancer has been described by few authors. In 1986, DeMuylder et al. [20] described three cases of uterine necrosis in over 939 patients (0.3%) treated with radiotherapy for cervical cancer; all of them required surgical management. In 2006, Marnitz et al. [21] described two cases of uterine necrosis; one patient with stage IIB cervical cancer had a partial response to the treatment but showed an early (6 days after completion of CCRT) combined necrosis of the uterus and bladder, which required laparotomic exenteration. In 2006, Micha et al. [22] described a case of cervical cancer treated by radical hysterectomy followed by CCRT that presented recurrent pelvic necrosis, which required multiple laparotomies over 15 years. Matthews et al. in 2007 [12] reported a case of complete uterine necrosis that presented five months following chemoradiation and was managed via abdominal hysterectomy; the histopathologic report showed a complete response to the treatment.

To date, Fawez et al. [23] have performed the most extensive case series of necrosis following chemoradiation. They reported five cases, all treated successfully with conservative management, based on smoking cessation, antibiotics, hydrogen peroxide vaginal douches, and opioids. Fawez et al. [23] suggested that surgical management should be avoided, and conservative treatment should represent the gold standard of care when possible/feasible. However, in cases like ours, or in cases where necrosis has associated hemorrhagic phenomena, necrosis opens improper anatomical spaces with communication between the vagina and pelvic cavity such as necrosis of vaginal fornixes, or the destruction of extensive areas of tissue or causing various types of fistulas, the surgical approach is indispensable.

In our case, conservative medical management was not feasible because of the acute and progressive increase of inflammatory indexes, the presence of sepsis, and the worsening clinical picture. Indeed, the first surgical intervention was finalized to the resolution of the emergency, and yet, as described above, consisted of removal of the necrotic tissue including the body of the uterus and adnexa, then carrying out supracervical-hysterectomy, bilateral salpingo-oophorectomy, and abscess drainage. On the other hand, the second surgery had a palliative therapeutic intent.

Radical hysterectomy after CCRT in LACC continues to fuel debate, since the instrumental evaluation of residual disease before surgery is uncertain. In fact, both MRI and PET/TC remain suboptimal in the evaluation of response to initial treatment because of poor specificity and high risk of false negative [24]. Of note, radical hysterectomy following CCRT in which an insufficient dose was provided, as in our case, represents a standard of care [25], and, in persistent bulky disease, may offer a significant benefit in terms of long-term survival [18,26].

## 4. Conclusions

Necrosis of pelvic tissues is a rare complication of CCRT in patients treated for locally advanced cervical cancer. Surgical management is recommended when laboratory tests, radiologic imaging, and clinical symptoms are not reassuring.

Thus, in patients that come to our attention after CCRT with persistent abdominal-pelvic pain, who have questionable findings in their imaging results, elevated inflammation parameters, or both, the differential diagnosis should include the suspicion of radiogenic necrosis of the uterus and other pelvic organs. Laparoscopy is an ideal technique to confirm or reject this diagnosis, in addition to allowing the performance of important therapeutic surgical steps. The present paper contributes to clarifying the central role of surgery, in particular when actinic necrosis determines complications such as abscess, fistulae, and extensive tissue destruction that cannot be medically conservatively managed.

## Figures and Tables

**Figure 1 diagnostics-12-00440-f001:**
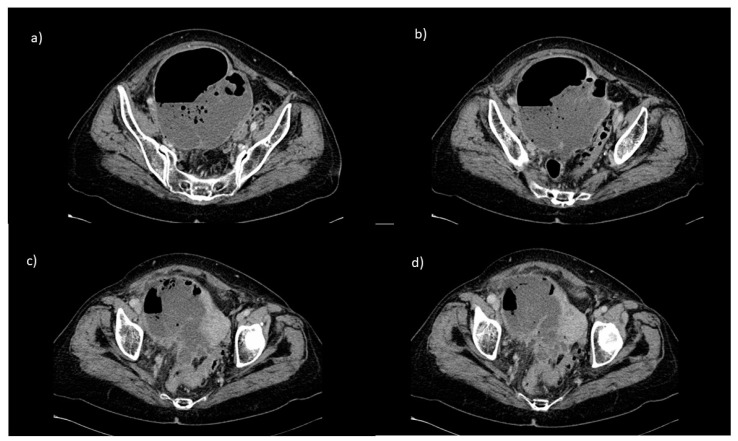
Preoperative CT scan showing the presence of a cervical tumor with extensive necrosis (78 × 43 mm), infiltrating the bladder and without cleavage plan with the rectum. Adjacent to the tumoral mass, a large (multichambered) abscess measuring 124 × 85 × 125 mm was observed which extended to the right parietocolic wall (**a**–**d**).

**Figure 2 diagnostics-12-00440-f002:**
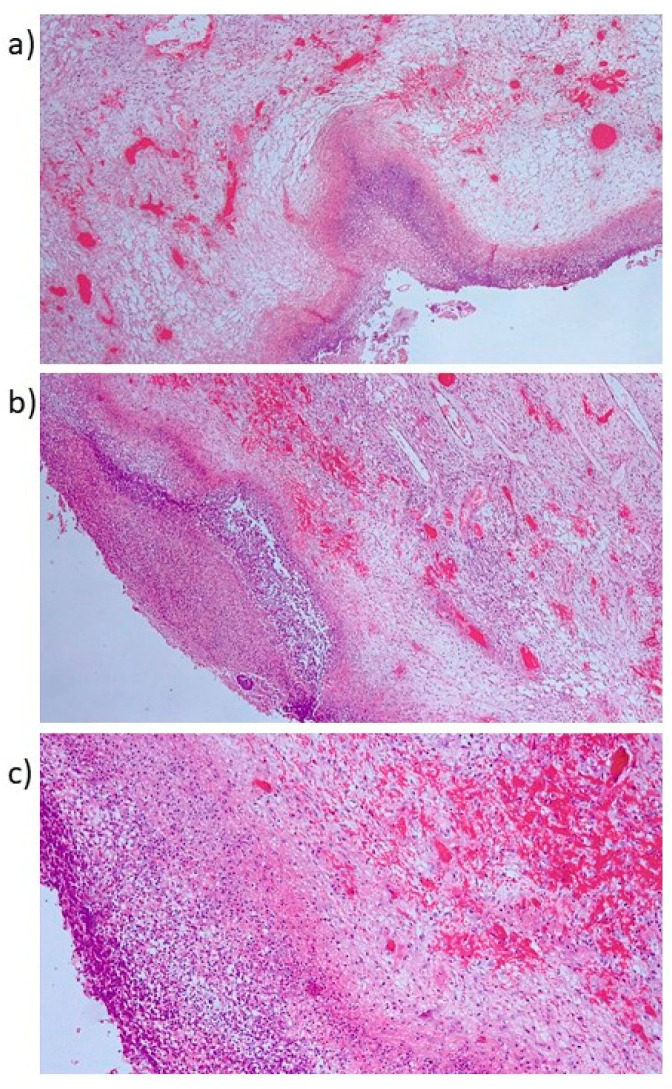
Definitive histopathological examination: (**a**,**b**) actinic necrosis in pelvic abscess (Hematoxylin-Eosin, HE, 100×); (**c**) magnified vision (200×) of necrotic debris and suppurated acute inflammation (HE).

## Data Availability

Original clinical, laboratory, and instrumental data can be found in the patient chart archived at the Department of Obstetrics and Gynecology and are available on request from the corresponding author.

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
