# Peer review of "Diagnostic Framework of Pelvic Massive Necrosis with Peritonitis following Chemoradiation for Locally Advanced Cervical Cancer: When Is the Surgery Not Demandable? A Case Report and Literature Review"

_diagnostics, 2022, doi:10.3390/diagnostics12020440_

Round 1

Reviewer 1 Report

The authors describe an interesting case of sepsis towards the end of treatment (CCRT+Brachy) for a very locally advanced cervical adenocarcinoma. While the case is interesting, I have some concerns

Major concern is that the authors tend to attribute the patient’s turbulent clinical course to the treatment and less so the severe extent of local disease. Certainly radiation therapy can cause necrosis within the pelvic structures, but as the others highlight, this is typically a late side effect of treatment. In this particular case, the patient had very locally advanced disease notable for involvement of the bladder and rectum. As such, fistulous communication between these organs was an expectation of therapeutic effect, rather a possible late adverse reaction. This scenario pre-disposed the patient to severe infection/abscess formation which likely precipitated, if not caused the majority of normal tissue necrosis.

Due to this, I feel like the discussion focuses too much on the acute/late effects of RT, rather the risks of treating patients with such advanced disease. I feel like a literature review detailing that topic would be much more relevant.  

Minor concerns:

-Lines 140-141 – Improper spacing

-Radical hysterectomy following CCRT in which an insufficient dose was not (or unlikely to be) provided is not controversial, rather standard of care.

Author Response

Point-by-point reply to Reviewers’ comments

Reviewer 1

The authors describe an interesting case of sepsis towards the end of treatment (CCRT+Brachy) for a very locally advanced cervical adenocarcinoma. While the case is interesting, I have some concerns.

Reply: Thank you for your comments. I revised accordingly to your indications. I hope that the revised version addresses your concerns.

-Major concern is that the authors tend to attribute the patient’s turbulent clinical course to the treatment and less so the severe extent of local disease. Certainly, radiation therapy can cause necrosis within the pelvic structures, but as the others highlight, this is typically a late side effect of treatment. In this particular case, the patient had very locally advanced disease notable for involvement of the bladder and rectum. As such, fistulous communication between these organs was an expectation of therapeutic effect, rather a possible late adverse reaction. This scenario pre-disposed the patient to severe infection/abscess formation which likely precipitated, if not caused the majority of normal tissue necrosis.

Reply: Thank you for your punctual and important comment that I would like to report in its entirety. Consistently, the following sentence have been added at lines 164-167 “In our particular case, the patient had a very locally advanced disease, notable for involvement of bladder and rectum, and, as such, fistulous communication between these organs was an expectation of therapeutic effect. This may have favored severe infection and abscess formation which likely precipitated the normal tissue necrosis.” Moreover, I have added the extent of disease among the risk factor for developing necrosis at lines 149-152, as follows: “Risk factors include total dose of radiation therapy, the radiation therapy field size (likely as function of disease extension), advanced stage disease, history of abdominal surgery (with compromised vascular supply), and diabetes.”

-Due to this, I feel like the discussion focuses too much on the acute/late effects of RT, rather the risks of treating patients with such advanced disease. I feel like a literature review detailing that topic would be much more relevant. 

Reply: Thank you for your comment and suggestion. According to your indication I have added a review of the main literature that detail how advanced stage disease cervical cancer, as our case, may be associated with increased risk of complications from CCRT and also surgery after CCRT. Thus, at lines 140-148 I have added the following: “…Moreover, of note, tumor extension at diagnosis have been correlated with in-creased incidence of complications, such as fistulae (recto-vaginal and vescico-vaginal fistulae), which represent a therapeutic challenge in very locally advanced cervical cancers [13-15]. Therefore, the use of CCRT in bulky and very advanced cervical cancer, as in our case, has major risk. In particular, it is often reported that bladder involvement and infiltration is associated with a strong fragility of vescical wall after radiation that favor almost constantly the complications related to radiotherapy with consequent fistulae formation [14,16,17]. Also, surgery after CCRT in such advanced stage of disease may be related to more frequent late complications [18].”

Minor concerns:

-Lines 140-141 – Improper spacing

Reply: I have corrected spacing.

-Radical hysterectomy following CCRT in which an insufficient dose was not (or unlikely to be) provided is not controversial, rather standard of care.

Reply: Thank you for your comment. I have added the following sentence: “Noteworthy, radical hysterectomy following CCRT in which an insufficient dose was provided, as in our case, represent a standard of care [25], and, in persistent bulky dis-ease, it may offer a significant benefit in term of long-term survival [18, 26].” (lines 234-237).

Reviewer 2 Report

Interesting case report, especially because therapeutic complications are less frequently published in the literature.

Please specify why the authors have preferred a ureterocutaneostomy instead a Bricker  type reservoir tailored from another part of the bowel (sigmoid or descendent colon, or jejunum).

Please describe in more details what was the therapy when the patient was re-hospitalized for recurrent multiple abscesses (no surgery, just drainage? or only antibiotics?)

Author Response

Point-by-point reply to Reviewers’ comments

Reviewer 2

Interesting case report, especially because therapeutic complications are less frequently published in the literature.

Reply: I appreciate your positive comments. I have revised the manuscript according to your suggestions.

-Please specify why the authors have preferred a ureterocutaneostomy instead a Bricker type reservoir tailored from another part of the bowel (sigmoid or descendent colon, or jejunum).

Reply: Thank you for your comment. It is a pleasure to notify you that typically we use the Bricker technique, as you can see from our works and in particular from the following “Madeddu C, Kotsonis P, Lavra F, Chiappe G, Melis L, Mura E, Scartozzi M, Macciò A. Next generation sequencing driven successful combined treatment with laparoscopic surgery and immunotherapy for relapsed stage IVB cervical and synchronous stage IV lung cancer. Oncotarget. 2019 Mar 12;10(21):2012-2021”. In the present case the reasons for the choice of performing ureterocutaneostomy have been indicated in the text at lines 97-98: “Because of the ileal resection, ileal conduit urinary diversion according to the Bricker technique was contraindicated; therefore, we opted for a left ureterocutaneostomy”.

Moreover, in accordance with the evidence reported in literature that indicates that ureterocutaneostomy may be considered preferable in cases of patients with significant comorbidities and history of intestinal radiotherapy, having been our patient septic with peritonitis and abdominal abscess, we opted for the ureterocutaneostomy instead of the Bricker technique. I have added the references supporting our choice in the text [Bulletin of Urooncology 2018;17:54-58; Review Int Braz J Urol. Jan-Feb 2022;48(1):18-30].

-Please describe in more details what was the therapy when the patient was re-hospitalized for recurrent multiple abscesses (no surgery, just drainage? or only antibiotics?)

Reply: I have added that the abscesses were treated with drainage and antibiotics therapy. See line 107: “…..that were treated with drainage and antibiotics therapy.”.

Round 2

Reviewer 1 Report

My concerns have been addressed

Author Response

Dear Reviewer

thank you for your positive comments. It is a pleasure to acknowledge that we addressed your concerns.